# Analysis and Detection of "Pink Slime" Websites in Social Media Posts

## ABSTRACT

Local news outlets play a vital role in providing trusted and relevant information to communities and addressing their specific needs and concerns. The emergence of news outlets posing as local sources and their spread on social media presents a significant challenge in the digital information landscape. This paper presents a comprehensive study investigating tweets featuring "Pink slime" news, which is a term that has been used to refer to these news outlets due to its deceptive nature. By analyzing a large dataset of tweets, we gain valuable insights into the patterns of these tweets and the origin of these tweets including automated accounts. We show in this work that extracting syntactical features proves valuable in developing a classification approach for detecting such tweets and show that the approach achieves 92.5% accuracy. We also show that our approach achieves near-perfect detection when grouping the tweets by URL.

ACM Reference Format:

Anonymous Author(s). 2024. Analysis and Detection of "Pink Slime" Websites in Social Media Posts. In *Proceedings of* . ACM, New York, NY, USA, 9 pages. https://doi.org/XXXXXXX.XXXXXXX

## 1 INTRODUCTION

During the 2016 United States presidential election and the emergence of the term *fake news*, Americans exhibited higher levels of trust in local news outlets compared to national ones [26]. The preference for local news demonstrated a belief that it offered a more trustworthy alternative to the perceived bias and sensationalism of national news [33]. This trust became a vulnerability that internal and external actors attempted to take advantage of to disseminate misinformation and shape public opinion. A report by the U.S. Senate Intelligence Committee in 2018 [27] found more than half a million tweets by external-operated Twitter accounts impersonating local news outlets during the 2016 election [26]. Exploiting public trust in local news began with the appearance of many news sites that posed as local news which have been labeled as "Pink slime" news [6].

"Pink slime," officially known as "lean finely textured beef", is an informal term used to describe a low-cost, processed beef product that is typically added to ground beef as a filler to reduce the overall fat [31]. However, the term has also been coined by journalists to refer to news outlets that appear to be local news, but in reality, they are not [44]. While Pink slime is not a formally recognized term, it serves as a practical depiction of these outlets in our research. Our

choice stems from the absence of an established, rigorously defined concept to name this particular phenomenon.

Although there have been sightings of these Pink slime news outlets since 2012, they have started to be more noticeable in the year ahead of the 2020 United States presidential election [7]. These media platforms adopt the names of cities and towns across every U.S. state, with almost no local reporters or physical newsrooms [40]. While those outlets claim to promote local journalism, their operations divert readership away from traditional local newspapers, raising concerns about their true intentions [38]. Similar to conventional news sources, Pink slime publications utilize social media for promotion. They often employ attention-grabbing headlines, commonly referred to as clickbait, to expand their presence on those platforms [4, 23]. Furthermore, some Pink slime outlets resort to automated bots and fake accounts to artificially boost their engagement statistics, making it more difficult for users to discern the authenticity of their content [11].

We present a novel study on the presence of Pink slime URLs in social media. Our study has four directions. First, we collect a large number of tweets, over 300K Pink slime tweets and 500K for national and local news, and curate them, e.g., we discard those with broken or expired URLs. Second, we study their textual organization by comparing the text of a tweet with that of the document (or news article) it references. This allows us to unearth interesting patterns of creation and dissemination of Pink slime tweets as compared to the rest of the news. For example, many of the Pink slime tweets copy the first sentence of the article it references. In addition, in most of those tweets, the sentences are cut short, at arbitrary positions. Since length is not an issue such a finding suggests that those tweets are generated by bots. Third, we aim to understand the factors– ranging from article tweet content to observed dynamics of user interaction with such tweets, like the number of likes– on detecting Pink slime tweets and explain which of those features contribute more toward detection. For instance, we observe that features related to the length or the time of the tweet are more useful that the interaction measures, such as the replies or the retweets. We show that we achieve 92.55% accuracy in detecting Pink slime tweets. Finally, in all the studies mentioned above, we contrast Pink slime tweets with the tweets that reference national and local news websites. Overall, this study advances our understanding of Pink slime news in the digital age and highlights the need for proactive interventions. By employing computational and analysis approaches, our paper makes the following contributions.

- We present a comprehensive analysis of tweets sharing URLs from Pink slime news websites, bridging the gap between journalism and data science.
- We utilize our analysis and insights into the sharing patterns within these tweets and develop a set of features tailored for tweets associated with Pink slime news.
- We develop a classification approach that detects tweets with URLs from Pink slime among tweets with news from

other sources. Our approach achieves 92.55% accuracy without relying on the textual content of the news article.

The paper is organized as follows: In Section 2, we discuss related work and news reports about Pink slime journalism. We define our problem in Section 3. In Section 4, we present the process of building and processing our dataset. We analyze tweets from our dataset and apply processing techniques from previous work in Section 5. We present our detection model and its results in Section 6. We discuss future directions and our conclusion in Sections 7 and 8.

## 2 BACKGROUND

Pink slime journalism is a relatively new phenomenon. Hence, it remains an understudied topic in academia compared to similar concepts, such as fake news, misinformation, and media bias [9, 20, 21, 24, 25]. During our investigation of this subject, we discovered that there is minimal research available in the literature that defines and examines the effects of Pink slime news. Large-scale language models, like ChatGPT, acknowledge the shortage of sufficient academic research (Figure 1) [28].

Numerous news articles and investigative reports have been published examining the notion of Pink slime journalism. These attempts aim to uncover the origin of these outlets and trace their dissemination [6, 22, 26, 44]. Alongside these news reports, a few research papers investigate the topic of Pink slime websites, examining two primary aspects: the website's attributes and its published content, as well as the consumption of their content [19, 23]. Furthermore, some studies investigate the characteristics of networks comprising multiple outlets rather than individual ones [32].

Bengani studies the history of the uprising of Pink slime outlets and their networks [6, 7]. Several investigative reports talk about how Pink slime outlets are more active during the times of elections [4, 7]. An article from The New York Times alleges that certain media outlets are being directed by political groups and corporate P.R. firms to continuously publish positive stories about a specific candidate or negative ones about their opponents [1]. Some local news outlets in swing states, such as Michigan or Pennsylvania, have reported the spread of Pink slime outlets in their states [14, 35].

Pink slime news outlets aim to compete with existing local news outlets in the same localities. Therefore, an important research problem is studying how they behave compared with the corresponding local establishments. The 2021 paper, by Karell and Agrawal, studied the content of 122,054 news articles from Pink slime websites and compared them with 90,689 news articles from the corresponding local news outlets. Their findings showed that the Pink slime websites have three distinct behaviors: publishing relevant online text and data, such as gasoline prices, public reports, or press releases; capitalizing on national political controversies; and utilizing emotional appeals to attract moderate readers and encourage further engagement [19]. A more recent work, by Moore et al., claims to present the first empirical analysis of individuals' engagement with Pink slime journalism. While the study reveals that a relatively small number of Americans visit Pink slime websites, the findings emphasize the significance of further research on this content type [23]. Their results indicate that over 9 million Americans accessed a Pink slime website during the 2020 election.

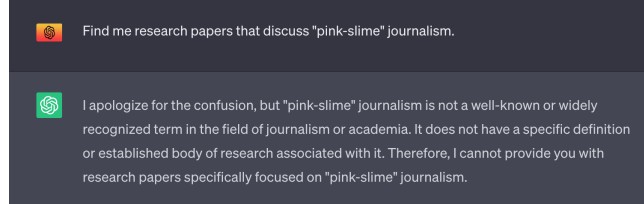

Figure 1: A screenshot showing the response of ChatGPT when asked about academic research on Pink slime.

The research by Royal and Napoli studies one of the largest network of pink slime outlets as a potential future modern model of local news reporting [32]. The finding of this study suggests that the network fails to adequately address the informational needs of the communities it claims to represent. These findings raise doubts about the effectiveness of its approach, which relies on generating automated content with minimal human-written stories to sustain a vast network of outlets nationwide. While local newspapers are facing financial challenges, this study highlights that automated, large-scale national operations are an inadequate substitute for the resource-intensive efforts of traditional local news. However, the model followed by such pink slime outlets may provide insights for those seeking to promote genuine local news in the future [32].

Overall, we notice a dearth of of systematic, quantitative, with empirical evidence research on phenomenon of Pink slime news. Hence, we believe that there are many opportunities for data-driven researchers on Pink slime journalism, its consumption, and circulation on social media platforms. Our work is motivated mainly by the lack of similar work on this problem from a computational perspective. We seek to uncover unique features and behaviors associated with tweets that share Pink slime URLs and compare those to tweets that share URLs of news articles from broadly recognized local and national outlets news. Understanding the distinctive attributes of those tweets is an important first step since it may enable social media platforms to create a more responsible environment by flagging posts that do not seek to inform, but rather seek to promote propaganda and persuasion [13, 41, 43].

## 3 PROBLEM

In this work, we focus on a particular study of tweets that share links from Pink slime websites, which we call *Pink slime tweets* throughout the paper. We identify two tasks: **First**, we seek to identify the distinguishing characteristics of tweets containing URLs from Pink slime news websites with a comparison to tweets with national and local news URLs. **Second**, by leveraging the findings and insights gained from our analysis we aim to show that we can find sufficient features for an efficient detection approach for these tweets. While one can create lists of pink slime websites and use them to detect such tweets, we aim toward a more comprehensive solution, which is able to tag such tweets for new, not previously seen pink slime websites. Such technology can prove valuable as it can track the emergence and vanishing of pink slime news outlets, a pattern often influenced by significant political and societal occurrences over time, like election and vaccination campaigns.

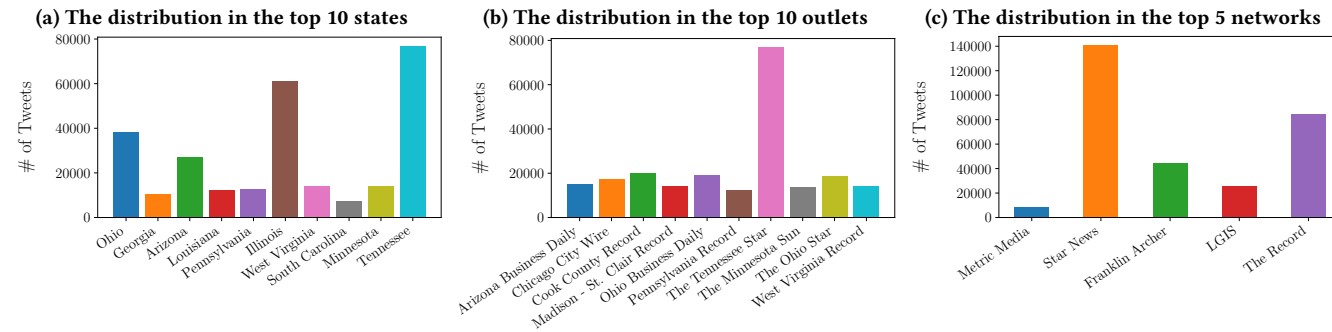

Figure 2: Distribution of Pink slime tweets in our datasets per states, outlets, and networks.

Table 1: The distribution of Pink slime outlets with respect to networks. The column *coverage* shows the number of outlets per type of local coverage (city-county-region-state)

| Network | outlets | states | coverage |
|---|---|---|---|
| Star News | 11 | 10 | (4-0-0-7) |
| The Record | 10 | 8 | (8-0-1-1) |
| LGIS | 36 | 1 | (33-3-0-0) |
| Metric Media | 989 | 50 | (530-141-271-47) |
| Local Report | 48 | 10 | (20-9-19-0) |
| Franklin Archer | 193 | 51 | (187-2-1-3) |
| Locality Labs | 16 | 2 | (16-0-0-0) |
| American Independent | 5 | 5 | (0-0-0-5) |
| American Catholic | 6 | 6 | (0-0-0-6) |

## 4 DATASET

In our paper, we experiment with tweets that contain news URLs from Pink slime, local and national news outlets. Therefore, we built a dataset with tweets in each category. We describe the process of collecting and curating the data in this section. The process has two main steps. First, we constructed a list of news outlets in each category. Second, we used Twitter API [1] to collect tweets that share URLs from each outlet in our list. We give the details below.

### 4.1 Collecting News Outlets

*4.1.1 Pink slime Outlets.* For the Pink slime category, we compiled a comprehensive list of Pink slime outlets. The list is built using data from several sources, including the Tow Center and The New York Times [2, 7, 40]. The list includes a total of 1,313 outlets that are spread across all 50 states and DC. Each of these outlets is associated with a particular network with a total of 8 networks. Table 1 shows more details about these networks. We also give the distribution for the top-10 states with the largest presence of pink slime outlets in Figure 3.

*4.1.2 Local and National Outlets.* Regarding national category outlets, we aim to make sure that the news outlets in our list represent different leanings in the political spectrum. Therefore, we use the Media Bias Chart from AllSides.com[2] and select 25 outlets from left, left-leaning, centrist, right-leaning, and right stances. In the

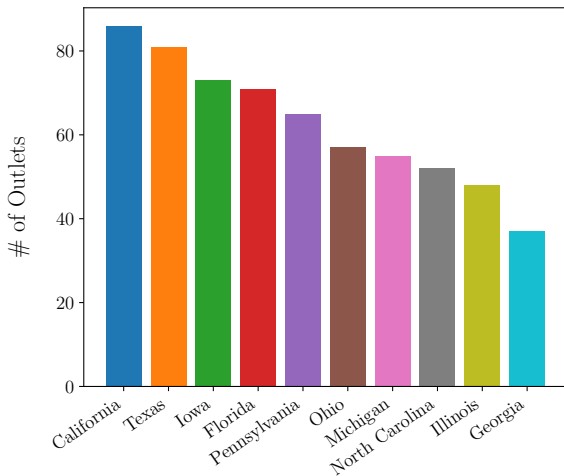

Figure 3: The distribution of Pink slime news for the top-10 states.

local news category, we first identify the localities where the Pink slime outlet operates, then we manually search in Google News for outlets using those locality keywords. We collect a list of 50 local news outlets, such as azcentral, Chicago Tribune, and PennLive.

### 4.2 Collecting Tweets with News URLs

*4.2.1 Pink slime Tweets.* After constructing the comprehensive list of outlets, the next step is that of gathering tweets that share URLs for news articles from each website in our list. To achieve this, we use the Twitter API to query tweets containing the prefix URL of the news website (e.g., www.newsoutlet.com). We collect tweets between January 2016 to May 2023. We download the text content of the news article from the URL included in each tweet. We collect 348k tweets with Pink slime URLs, but after cleaning only 305k contain accessible URLs. Figure 2 shows the distribution of these tweets per network, top 10 outlets, and top 10 states.

Upon investigating the temporal distribution of the collected tweets, we noticed a surge in the volume of Pink slime tweets between May 2022 and September 2022. This temporal trend is shown in Figure 4, which illustrates the overall distribution and the distribution of tweets in the originating outlets, networks, and states. This surge is attributed primarily to three outlets: Arizona Business Daily, Ohio Business Daily, and Chicago City Wire. The majority of

[1]https://api.twitter.com
[2]https://www.allsides.com/media-bias/media-bias-chart

the tweets from those outlets, precisely 95.82%, 99.90%, and 82.68%, were exclusively concentrated in this five-month window. Moreover, our investigation revealed that 87.17%, 99.87%, and 80.41% of these tweets originated from an account that belonged to the respective outlets.

*4.2.2 National and Local Tweets.* We collect tweets from both national and local news outlets, employing the same time frame utilized in our Pink slime tweet collection. Specifically, we build a dataset comprising of 500k tweets from national news and an equivalent number from local news outlets. We also download the URLs included in those tweets. This dataset is used in our subsequent experiments and in-depth analyses.

## 5 ANALYZING WRITING PATTERNS

Following our data collection, we manually examined a sample of 1000 Pink slime tweets from our dataset to explore potential directions for analysis. Throughout our inspection, we found instances where the tweets were posted by accounts that belonged to news outlets, others that appeared to be posted by bot accounts, and others by ordinary users. We believe that this observation applies to all pink slime tweets. However, what interested us more was the common behavior of these accounts, specifically their composing and editing practices in Pink slime tweets. For example, we noticed that Pink slime tweets exhibit less discussion and contextual information compared to other news tweets. For example, we came across many instances where tweets featured incomplete sentences that were copied from the shared news article and subsequently truncated (Figure 5). In this example, the title of the shared news article is copied and cut short. This action was not deemed necessary due to the length restrictions of tweets. In other examples, the truncated text is a quote from within the body of the article. Following these findings, we sought to explore the relationship between the text of the tweet and the text of the Pink slime news article in more systematically.

### 5.1 Relationships Between Tweet and News Articles

There are many works in the literature that analyze and study tweets text, but we found few that focus on tweets with news URLs or URLs, in general, [10, 16, 37, 42]. One work analyzes the changes to the news article headlines after being shared on Twitter and shows that these edits differ from one news outlet to another [17]. Another work studies general hyperlinked tweets indicates that not all tweets containing URLs are composed in the same way [3]. Their findings show that some tweets simply copy the title of the URL, while others include quotes from the news article. Some may contain additional commentary or text written by the user, while others may include multiple elements such as hashtags, images, or mentions of other users [3]. These findings apply to all tweets with URLs, including news URLs and more importantly Pink slime URLs. The nature of such tweets can have an impact on how convincing they appear to other users and how likely they are to be shared or spread further. By analyzing the text of the tweet, such as whether it contains only the title of the article or additional commentary, researchers can gain insights into the motivations, bias, or stance of the users who share these links [36]. For instance, a potential

study could explore whether tweets featuring the news article's title are more likely to be posted by automated bots, whereas tweets including excerpts from the article or user-generated content have a higher likelihood of human sharing. Next, we show the results and implications of applying the method of segmentation [3] on the tweets in our dataset.

### 5.2 Segmentation of the tweet text

To investigate the *editing behaviors* performed on tweets with Pink slime news URLs, we use the segmentation method for tweets with URLs, in which we inspect the similarity between the text of the tweet and the title and text content of the shared URL [3]. To achieve that, we follow the method proposed by Aljebreen et al., which is summarized as follows: (1) we check if the title of the news article is copied into the tweet; if yes then we mark it as a title segment or *ttl*. (2) For the remaining, unsegmented, text of the tweet we check if it contains quotes copied from the main body of the article. If yes, we mark them as body segments, or *bdy*. (3) The remaining parts of the tweet are assumed to be added by the user and, hence, are marked as user segments, or *usr*. We followed the algorithm described in [3], which mostly follows known string matching strategies. We complement it with several heuristics to take into account minor edits to the title and body when copied to the tweet, such as adding hashtags or user handles or shortening the copied text from the news article.

*5.2.1 Setup.* In order to study the distinctive characteristics of Pink slime tweets in comparison to other news tweets, we conducted segmentation across all three datasets: Pink slime, national, and local news tweets. We use all 305k collected Pink slime tweets. As for national and local news tweets, we sampled the same number of tweets from each dataset (Section 4.2.2) while adhering to the same time interval and tweet frequency as the Pink slime dataset.

*5.2.2 Results.* In the segmentation process, each processed tweet is assigned a pattern based on its segments. For instance, a tweet with the pattern: (*ttl*) consists only of a title segment, while another with (*bdyusr*) pattern has a quote from the body of the article followed by user-added text. The results reported by [3] indicate that the most common 3 patterns are *ttl*, **usr**, and *bdy*. In our experiments, the observed distribution is different from theirs and the patterns differs in each dataset. In Figure 8, we show the most common 7 segmentation patterns. The results reveal distinct patterns in Pink slime tweets as compared to national and local news tweets. Pink slime tweets predominantly exhibit *ttl* pattern, accounting for over 74.36%, whereas this pattern is observed in only 20.49% and 24.2% of national and local news tweets, respectively. On the other hand, the most prevalent pattern in national and local news tweets is *usr*, present in 35.64% and 32.29% respectively, whereas Pink slime tweets exhibit this pattern in only about 9.66% of cases.

The overall results give a clear indication of the posting behavior of Pink slime tweets as a population: they include little user text. Moreover, the segmentation distribution differs between some outlets. In Figure 6, we show the segmentation distribution of Pink slime tweets among different networks, outlets, and states. We will show later, in Section 6, that these patterns identified in the tweets with news URLs, across the three categories, can also serve as useful

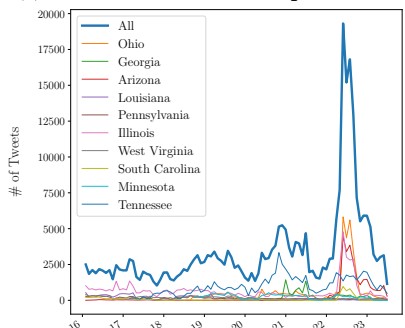
(a) The distribution in the top 10 states

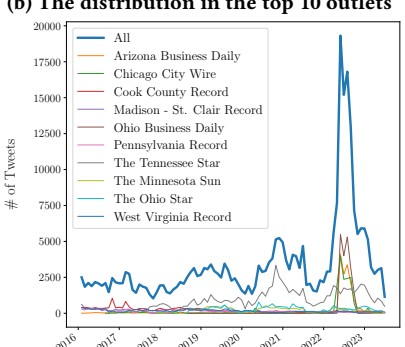
(b) The distribution in the top 10 outlets

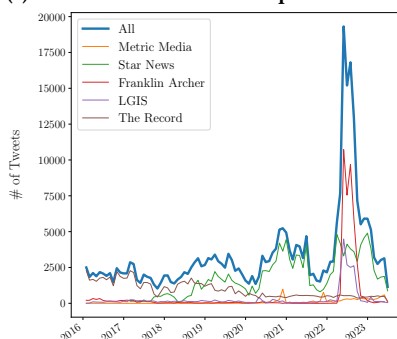
(c) The distribution in the top 5 networks

Figure 4: Temporal distribution of Pink slime tweets in our datasets per states, outlets, and networks.

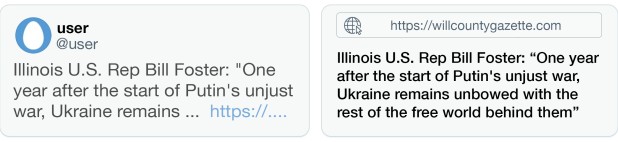

**Figure 5: An example of an excerpt is copied from the news article and abruptly truncated.**

features for a machine-learning classification approach designed to detect such content. By analyzing a large dataset of tweets, an ML model can learn to identify common patterns that are associated with Pink slime news URLs, and the dominant features are related to the presence or lack of user-generated text.

## 5.3 Special Cases of Segmentation

Once we have reached the results presented in this section, which reveal the predominant editing patterns of the tweets in each category, our focus shifts towards identifying more specific markers within the range of these patterns. This is particularly crucial in the context of our primary subject of investigation, which is the Pink slime tweets. In the following sections, we delve into three noteworthy instances where we can discern additional features associated with these patterns.

*5.3.1 Outlet reference.* One of the most common patterns is when the title appears in the tweet followed by a segment that is assumed to be added by the user. We notice that the nature of *usr* segments differs in length and textual content. In many instances, the tweets refer to the source of the shared news article by mentioning its user handle in Twitter, e.g. *via @BreakingNews*. An example of this tweeting behavior is shown in Figure 7. We further search the datasets for these patterns and we find that it is more common in the national and local tweets than in the Pink slime tweets. Around 49% of the national news tweets with *ttl-usr* pattern contain this case, while only 21% from the local tweets and 14% of the Pink slime tweets have them.

*5.3.2 First sentence quote.* Another observed case is the location of the quoted text, the *bdy* segments, in the news article itself. This was brought to our attention by noticing the presence of many tweets with incomplete quotes in Pink slime tweets, which appear to be copied from the beginning of the news article. We show two examples, in Figure 10, from both local (top tweet) and Pink slime (bottom tweet) datasets. We study tweets with *bdy* pattern in all datasets and attempt to determine the overall prevalence of this copying pattern. Our analysis finds this pattern in 57% of Pink slime news tweets with *bdy*, 27% of local tweets, and less than 1% of national news tweets.

*5.3.3 Paraphrased title.* We observed that certain tweets with *usr* segments, often presented a paraphrased version of the article's title instead of an addition or discussion by the user. For example, the tweet shown in Figure 9 is marked to have *usr* segments. However, the title of the news article appears to be paraphrased and copied into the tweet. To confirm and test these observations on a larger level, we employed the GPT3 language model [28] to detect the paraphrasing of the title. For each tweet, we passed both the *usr* segment and the title and asked GPT3 to determine if the two pieces of text are paraphrased of each other. However, upon manually reviewing a sample of 100 tweets that we processed with GPT3, we discovered numerous instances of inaccuracies and false positives. Among this sample, we found 21 tweets to be false negative and 44 were false positive. These findings highlighted the limitations of relying solely on automated methods in such problems.

## 6 DETECTING PINK SLIME TWEETS

The findings of our analysis in the previous section provide us with an opportunity to look for a set of features that can be used to develop a classification algorithm for Pink slime tweets. In addition, our goal is to identify the feature subset that is the most important for the detection of tweets with Pink slime URLs.

### 6.1 Methodology

The use of the Random Forest machine learning model presents a compelling approach for classifying tweets containing Pink slime news. Random Forest is a powerful ensemble learning technique that combines multiple decision trees to make accurate predictions [12]. Its ability to handle high-dimensional data and capture complex interactions among features makes it well-suited for the task of classifying tweets based on the presence of Pink slime news [8, 29]. By leveraging the ensemble nature of Random Forest, we can effectively harness the collective wisdom of multiple decision trees to

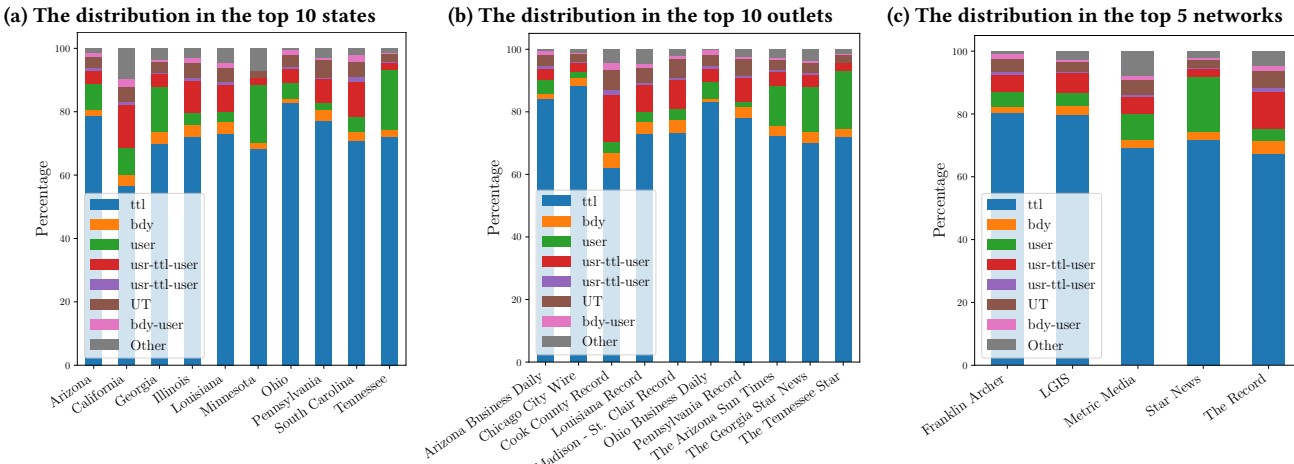

**Figure 6: The observed segmentation patterns for Pink slime tweets. We organize them by state, outlet, and network.**

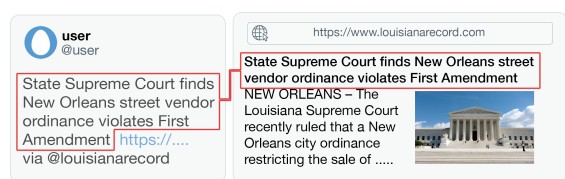

**Figure 7: An example tweet where the title is followed by mention reference to the source of the URL.**

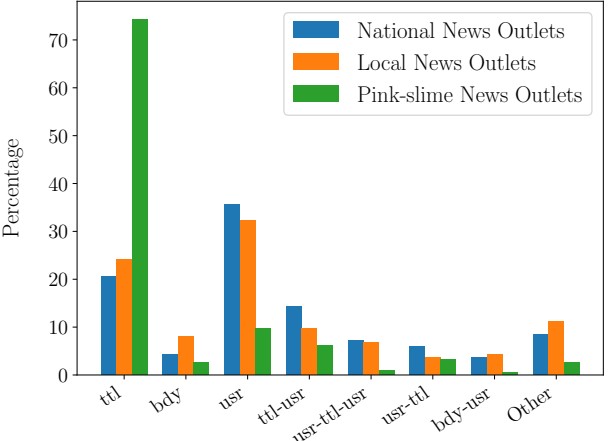

**Figure 8: A comparison of each dataset's most common segmentation patterns.**

improve the overall classification performance. Additionally, the model's interpretability, compared to deep neural networks [5], allows us to gain insights into the key features driving the classification, enabling us to understand the distinguishing characteristics of Pink slime news tweets [39].

## 6.2 Features

In our experiments, we try different combinations of features and attempt to find the most useful ones for detecting Pink slime tweets

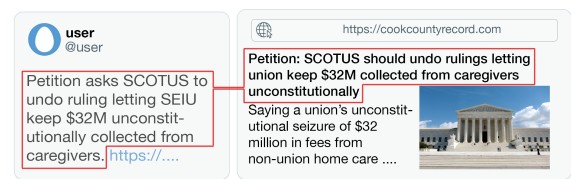

**Figure 9: An example of a Pink slime tweet where a paraphrased version of the URL's title appears.**

among other news tweets. In addition to the segmentation signals, we discussed in the previous section, we extracted additional features from the tweets. We organize the features into six groups: Segmentation, Other patterns, Special elements, Interaction, Structural, Temporal, and Misc features. Table 2 summarizes these feature sets.

**Segmentation features** In the segmentation features, we use the patterns of the results when we process the tweets using the segmentation patterns, such as *ttl-usr, bdy, usr* ... etc. We converted this nominal feature into multiple binary ones. For example, 1 if *ttl-usr* appears and 0 otherwise.

**Other patterns features** Here we use the observations we discussed in Section 5.3 and extract two features related to them and the tweet patterns. In addition, we add a within tweet location feature of a URL. A tweet may appear at the beginning or end of a tweet, or somewhere in the middle of it.

**Special elements features** Hashtags, mentions, and emojis play significant roles in tweets. Hence, we transform their presence into distinct features for analysis, such as the number of hashtags and their length.

**Interaction features** One of the important parameters that we collected during the gathering of the tweets was *public metrics* which is a count of the engagement interactions that the tweet receives in real-time, which are Replies, Likes, Retweets, and Bookmarks. We use these interaction metrics as features for our tweets.

**Structural features** Other textual properties were extracted during the processing of the tweets. These properties include the

**Figure 10: Two examples of tweets, the top one is a Pink slime tweet and the bottom is a local news tweet, where the first sentence of the article is copied into the tweet.**

length of the tweet, the number of sentences, and the number of characters in the tweet that are punctuations.

**Temporal features** Includes two features that are related to the tweeting time, we focus on the time of the day the tweets were posted. Also, we use the day of the week as another temporal feature.

**Misc features** Lastly, we incorporate additional metrics that we collected from the Twitter API. The first metric, *annotation_count*, is the number of mentioned entities recognized in the tweet. The second one, *geo_info*, returns the geographical location of the tweet's origin.

We do not include www.newsoutlet.com as features. The reason is that we want the model to learn features that are independent of the name of an outlet. Such a model will be more robust to identifying unseen pink slime tweets.

## 6.3 Experimental Setup

In our experimental study, we utilize a cross-validation approach to assess the RF model performance. The tweets in our datasets are randomly divided into five folds based on the outlets they belong to. Four of these folds constitute the training set, while the fifth fold is reserved for testing. We create a model based on the training set's features and then apply this model to the unseen testing set. This process is repeated five times, each time designating a different fold for testing. Our reported performance metrics are based on the average results obtained through this approach. Notably, this evaluation includes the application of a zero-shot detection technique, where no Pink-slime outlet is used during both the training and testing phases.

*6.3.1 Evaluation metrics.* In evaluating our technique, we rely on essential metrics like *F1* score and *accuracy*. The *F1* score offers a balanced assessment of precision and recall, crucial for evaluating the model's ability to correctly identify Pink slime tweets while minimizing false positives. On the other hand, the *accuracy* measure provides an overall measure of correctness in tweet classification and shows the proportion of all correctly detected tweets.

**Table 2: The 6 groups of features utilized in the pink slime tweet detection experiments.**

| Feature | Description |
|---|---|
| **A) Segmentation features** | |
| tweet_pattern | (*T, B, BU* ...) the segmentation pattern |
| pattern_length | (0-...) number of segments in the pattern |
| ttl_presence | (0,1) whether *Title* segment is present |
| bdy_presence | (0,1) whether *Body* segment is present |
| usr_presence | (0,1) whether *User* segment is present |
| **B) Other patterns features** | |
| outlet_reference | (0,1) whether it has a reference to the outlet after the title, e.g. via @News |
| first_sentence_quote | (0,1) whether it has a direct quote the first sentence in the article |
| URL_location | (0,1,2) location of the URL in the tweet |
| **C) Special elements features** | |
| hashtags_count | (0-...) the number of hashtags |
| hashtags_length | (0-...) the length of hashtags in characters |
| mentions_count | (0-...) the number of user mentions |
| emojis_count | (0-...) the number of emojis |
| **D) Interaction features** | |
| retweet_count | (0-...) the number of retweets |
| reply_count | (0-...) the number of replies to the tweet |
| quote_count | (0-...) the number of times the tweet was quoted |
| like_count | (0-...) the number of likes |
| bookmark_count | (0-...) the number of times the tweet was was bookmarked |
| **E) Structural features** | |
| tweet_length | (0-...) the length of the tweet |
| sentence_count | (0-...) the number of sentences |
| punctuation_count | (0-...) the number of punctuation in the tweet, e.g. *!?,*. ...etc |
| **F) Temporal features** | |
| hour_of_day | (0-23) the hour of the day the tweet was posted |
| day_of_week | (0-6) the day of the week the tweet was posted |
| **G) Misc. features** | |
| annotation_count | (0-...) the number of annotations |
| geo_info | (0-1) whether the geo info is available in the tweet |

*6.3.2 Hyperparameters.* Tuning hyperparameters plays a critical role in enhancing the performance of Random Forest models. Through *random search cross-validation* which performs small-scale *k*-fold experiments we randomly experiment with different combinations of hyperparameters to optimize the model's performance. The parameters used in our model and our choice are as follows: the number of decision trees in the forest ($num_trees \in [300, 500, 700]$) and the maximum depth of these trees ($max_depth \in [10, 20, 30]$).

**Table 3: The results for identifying dominant features when only using local, national, and both settings with Random Forest.**

| Datasets | | ALL | ALL-{A,B} | ALL-{C,D} | {A,B} | {C,D} | {E,F,G} | {A,B,E,G} |
|---|---|---|---|---|---|---|---|---|
| Pink slime vs All news | Acc. | **0.93** | 0.79 | 0.90 | 0.77 | 0.70 | 0.82 | **0.91** |
| | F1 | **0.80** | 0.52 | 0.75 | 0.51 | 0.45 | 0.64 | **0.77** |
| Pink slime vs national | Acc. | **0.92** | 0.78 | **0.91** | 0.77 | 0.68 | 0.85 | 0.89 |
| | F1 | **0.75** | 0.58 | **0.76** | 0.50 | 0.44 | 0.68 | 0.74 |
| Pink slime vs local | Acc. | **0.92** | 0.83 | 0.87 | 0.78 | 0.69 | 0.82 | **0.91** |
| | F1 | **0.76** | 0.62 | 0.69 | 0.56 | 0.44 | 0.65 | **0.77** |

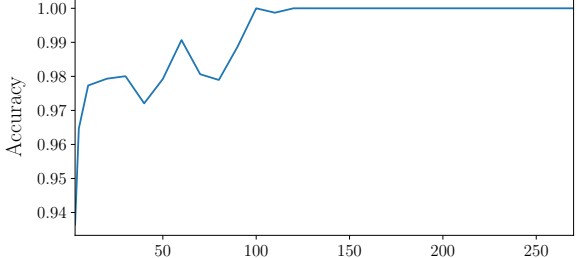

**Figure 11: The increase in the accuracy when the number of common URLs increases. The collective detection label is always correct after 100 URLs.**

### 6.4 Results

We trained our RF model to detect Pink slime tweets in three distinct settings, compared to national news, local news, and all news. Our model exhibits an accuracy of 0.926 and an F1 score of 0.8 when trained with all news. The model performs similarly when analyzing solely local news. However, in comparison to national news, it displays a slightly lower accuracy (Table 3).

### 6.5 Identifying Dominant Features

In Table 3, we show the results of our ablation study, where we aim to discover the most valuable set of features. Our results demonstrate that a combination of segmentation, structural, and annotation features deliver performance results most closely to the performance of the entire set of features. On the other hand, features that related tweet elements, such as hashtags, mentions, or emojis, show to be less useful.

### 6.6 Collective Detection by URLs

To enhance the precision of detecting individual tweets, we employ a grouping strategy. This strategy involves grouping tweets that share the same URL, and subsequently, we designate the majority label within each group as the collective label for all tweets associated with that URL. We resolve any tie by randomly assigning labels to the entire group. Our assumption is that tweets in our dataset tend to have similar attributes as a group. Therefore, this grouping process should boost the accuracy of detecting Pink slime URLs in individual tweets. Also, it should simplify the classification task by providing a single, collective label for groups of tweets that share identical URLs.

To assess the grouping concept, we examine subsets of the evaluation dataset, each consisting of tweet groups with equal sizes and shared URLs. In Figure 11, we show the increased accuracy as we have more URLs in a group. The accuracy reaches 100% around 100 tweets per URL. We believe that this is an interesting finding as it shows that tweets in isolation do not provide sufficient opportunity for learning, but collectively they do. Such observation is encountered in other works that study social media such as named entity recognition, entity linking, and misinformation [15, 18, 30, 34].

## 7 FUTURE WORK

The emergence of Pink slime news websites should motivate researchers to study not just this type of news, but to monitor its evolution and look for any new kinds of news that might appear in the future, especially in social media. As for our current problem, there are several aspects of Pink slime news we did not investigate. To gain a deeper understanding of the evolving nature of Pink slime news and its impact over time, future research can focus on the user aspect of these tweets. Studying the user accounts that post URLs from Pink slime outlets or engaging with them can provide a further understanding of the extent of reach these outlets have in social media. Another promising avenue for future research is to study tweets that are replies to Pink slime tweets and the content of the conversation around them [36]. Most Pink slime outlets do not have comment sections on their website, and these replies can be valuable substitutes for comments. Further exploration of the content of Pink slime tweets and their accompanying replies may look into deeper text mining, such as sentiment or stance detection. Such analysis can be helpful, especially when looking at tweets about controversial topics within public discourse. This extended analysis has the potential to enhance our understanding of the nuances and impact of Pink slime tweets in shaping online conversations.

## 8 CONCLUSION

In conclusion, our study unveils the distinctive nature of tweets with URLs from Pink slime outlets in contrast to national and local news tweets. We identify unique patterns derived from the user behavior when sharing Pink slime tweets. We employ these valuable insights into extracting an extensive set of features and developing effective machine learning models to detect Pink slime tweets, ultimately aiding in the ongoing effort towards a more secure and safe social media ecosystems. Our detection approach achieves an accuracy of 92.55% and an F1 score of 0.80. Moreover, we show that taking advantage of multiple tweets with the same URL can improve our results to reach a perfect prediction of Pink-slime tweets. Our work contributes to the broader dialogue on evolving information dissemination and equips researchers with essential tools to study this issue.

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

//api.semanticscholar.org/CorpusID:22615903

