# OpenReview forum: "Analysis and Detection of "Pink Slime" Websites in Social Media Posts"
_ACM.org/TheWebConf/2024/Conference — TheWebConf24_

### Official Review · Reviewer_FrKK · 2023-11-17

**Novelty:** 5
**Technical Quality:** 5

**Review:**

This paper presents interesting exploratory work on an emerging issue in communication and journalism studies; that of the "Pink Slime" phenomenon. The authors establish the dangers of Pink Slime as fake news outlets posing as legitimate sources that divert viewership/readership away from trusted news media. The remainder of the paper includes categorizing syntactical features of Pink-slime tweets that the authors then use to train a machine learning model. Though the premise of this paper is interesting enough, the quality and clarity of this paper was somewhat lacking. To start, the smallest issue being that the general formatting of figures and tables felt disorganized and difficult to follow. And there were many grammar and syntax errors that made passages difficult to read. As well as ill defined concepts and awkward sentence structures that made the general project narrative also difficult to follow, for example what is a Pink Slime Network? This seems to be an important feature of the data collection and descriptive summary but it is never solidly defined.

Also, there were issues establishing the relevance of this project amongst existing literature. In page two the authors start by saying there is
"minimal research available" on Pink-slime and in the same page go on to say they "notice a dearth of systematic, quantitative with empirical evidence research on the phenomenon of Pink Slime news". The authors also reference the fact that chatGPT did not have any references for journal articles that study pink slime journalism and include a screenshot of ChatGPT's of their query. This seems like an unnecessary misstep considering that ChatGPT cannot directly browse the internet in real-time (as of now). And I put in the same query on ChatGPT I received a very different answer on this version of ChatGPT. The biggest difference being that the ChatGPT response referenced by the authors did not even have a definition for pink slime, whereas my query did yield a definition as well as a number of sources to start searching for publications on pink slime. The difference in responses goes to show relying ChatGPT as an assessment of progress on a specific subject in a specific field is inadequate. Furthermore, I believe the authors standout contribution to the research on pink slime news is that they will be using a computational perspective. However, this brings me to my next criticism regarding their computational approach. They train a random forest classifier to detect Pink-slime tweets. However, though they justify the use of a random forest classifier, using just one classifier for exploratory work such as this feels lacking. I reference Abad, Gholamy and Aslani's (2023) study "Classification of Malicious URLs Using Machine Learning" which use K-Nearest Neighbor's, Support Vector's and DT alongside Random Forest to identify malicious urls. This study might have been very recent relative to when the authors wrote their project but there is other cited work on this subject. And many use multiple models to establish the credibility of specified features in classification tasks. The authors would benefit from testing these same pink-slime tweet characteristics across multiple models, especially after establishing which features perform the best for this type of classification task. Overall, though this is an interesting topic that uses a creative approach. But I recommend that authors develop their project a little bit more in terms of narrative structure and methodological execution.

**Questions:**

How did you arrive at the 1000 sample for the segmentation analysis? Is this an appropriately representative sample?

What is the us of generative AI's such as GPT3 in terms of tweet segmentation? I'm confused of the purpose of section 5.3.3. please clarify.

Regarding the segmentation features for the Pink-slime tweet random forest classifiers, if Pink-slime tweets overwhelmingly just feature the ttl segment why not have a binary of whether a tweet just include ttl or not?

What are the features of Pink-slime urls? Since these urls are linked to pink-slime news websites wouldn't investigating the features of the urls themselves be more telling in terms of identifying pink slime tweets? You mention that you don't include the names of websites because you want to classifier to be more generalizable but what about other features like the presence of an "html" or "php". How does pink slime url anatomy differ from urls from legitimate websites?

**Ethics Review Description:**

No ethics violations detected

**Reviewer Confidence:**

4: The reviewer is certain that the evaluation is correct and very familiar with the relevant literature

**Scope:**

3: The work is somewhat relevant to the Web and to the track, and is of narrow interest to a sub-community

---

### Official Review · Reviewer_WWcx · 2023-11-22

**Novelty:** 4
**Technical Quality:** 4

**Review:**

The paper delves into the phenomenon of "Pink slime" websites—deceptive outlets posing as local news—and specifically examines tweets sharing URLs from these sites. The authors explore the distinct writing styles of Pink slime tweets compared to (real) national and local news tweets. The study comprises four key components: tweet curation, textual organization analysis, detection factors, and the development of a classification model with an impressive 92.55% accuracy.

Strengths:
- S1. The paper highlights the societal risk posed by Pink slime sites, emphasizing the importance of studying their social media presence.
- S2. The authors collect news articles from Pink slime websites, providing a valuable dataset for the research community.
- S3. The study reveals clear differences in segmentation strategies between Pink slime tweets and real news tweets, helping to build a highly effective model to classify Pink slime tweets.

Weaknesses:
- W1. Skewed dataset: The dataset appears potentially skewed, with a substantial portion dominated by a few top websites (it seems that one website, The Tennessee Star' covers 25% of the data, and the top 10 sites cover about 75% of the tweet data). This raises concerns about the generalizability of observed patterns to all Pink slime tweets, as the writing styles may be primarily influenced by these major outlets.
- W2. Experiment design: The paper's experimental design lacks consideration for the temporal dynamics of the news industry. Conducting experiments on data split by time would provide more practical insights, especially given the evolving nature of news content.
- W3. Comparison with existing studies: The paper does not adequately address how Pink slime tweets differ from other alternative news media or clickbait news sites. Additionally, the authors do not explore the potential utility of existing methods developed for similar purposes.
- W4. Unclear motivation for classifier: The motivation behind building a classifier for Pink slime tweets is unclear, and the paper does not convincingly establish whether these tweets are harmful or engaging. A clearer justification for the classifier's practical implications would enhance the paper's overall impact.

**Questions:**

NA

**Reviewer Confidence:**

4: The reviewer is certain that the evaluation is correct and very familiar with the relevant literature

**Scope:**

4: The work is relevant to the Web and to the track, and is of broad interest to the community

---

### Official Review · Reviewer_jkko · 2023-11-23

**Novelty:** 5
**Technical Quality:** 4

**Review:**

In the study, a Pink smile dataset, including Pink simle outlets and national or local news tweets, is established and analyzed. Then 6 groups of features are extracted and a ranodm forest classifer is trained to classify pink sime tweets and the ordinary news tweets, achieves an accuracy of 92.55%. The study is a bit of fun.

However, the study may be improved in the following ways.
1. This seems the first work which studies Pink simle tweet identification. Thus, the constructed dataset had better to be published with the paper.
2. A tranditional classfication method RF in machine learning is used to identify the Pink smile tweets under the help of human curated features. What if we use a deep learning method like BERT to do this classification task without using these human curated features.

**Questions:**

The problems 1 and 2 can be answered.

**Reviewer Confidence:**

3: The reviewer is confident but not certain that the evaluation is correct

**Scope:**

4: The work is relevant to the Web and to the track, and is of broad interest to the community

---

### Official Review · Reviewer_qqmi · 2023-11-24

**Novelty:** 4
**Technical Quality:** 4

**Review:**

This research article investigates tweets featuring “Pink slime” news. Exploiting public trust in local news began with the appearance of many news sites that posted local news which has been labeled as “Pink slime” news. This study has four directions: (i) Collection of tweets over 300K Pink Slime tweets and 500 K for national and local news and curates them (ii) Textual organization by comparing the text of a tweet with that of the document (or news article) it references. (iii)  Understanding the factors ranging from article tweet content to number of likes (iv) contrasting pink slime tweets with the tweets that reference national and local news websites. The main objective of this paper is to present a comprehensive analysis of tweets sharing URIs from Pink Slime news websites bridging the gap between journalism and data science, and develop a classification approach to identify tweets with URLs from Pink Slime among tweets with news from other resources.

Strengths:
S1: The problem studied in this research work is quite interesting.
S2: The novelty of the work is well explained and the contribution of the work is explained in good manner.

Suggestion:
1) Apply deep learning models for classification.

**Questions:**

No

**Reviewer Confidence:**

3: The reviewer is confident but not certain that the evaluation is correct

**Scope:**

3: The work is somewhat relevant to the Web and to the track, and is of narrow interest to a sub-community

---

### Decision · Program_Chairs · 2024-01-22

**Decision:**

Accept

**Comment:**

The paper presents an intriguing exploration into the spread of pink slime news on social media, developing a classifier to identify tweets containing such news. This offers valuable insights into the contrast between traditional news and pink slime news. Reviewers concur on the importance of the topic and the usefulness of the analysis. However, there are areas for improvement: the absence of deep learning models and other classifiers that might enhance performance, and concerns regarding the methodology for literature search, particularly the use of ChatGPT for identifying relevant publications. Despite these issues, I believe the strengths of the paper, particularly its contribution to a novel and significant topic, outweigh its limitations.